# A Mediation Analysis of the Association between Fundamental Motor Skills and Physical Activity during Middle Childhood

**DOI:** 10.3390/children8020064

**Published:** 2021-01-20

**Authors:** Xiangli Gu, Priscila M. Tamplain, Weiyun Chen, Tao Zhang, M. Jean Keller, Jing Wang

**Affiliations:** 1Department of Kinesiology, University of Texas at Arlington, Arlington, TX 76019, USA; priscila.tamplain@uta.edu; 2School of Kinesiology, University of Michigan, Ann Arbor, MI 48109, USA; chenwy@umich.edu; 3Department of Kinesiology, Health Promotion and Recreation, University of North Texas, Denton, TX 76201, USA; Tao.zhang@unt.edu (T.Z.); Jean.keller@unt.edu (M.J.K.); 4College of Nursing and Health Innovation, University of Texas at Arlington, Arlington, TX 76019, USA; Jing.wang@uta.edu

**Keywords:** motor competence, obesity, physical fitness, physical activity, middle childhood

## Abstract

The purposes of the study were: (1) to investigate the associations between fundamental motor skills (FMS), health-related fitness (HRF) and physical activity (PA) during middle childhood; and (2) to examine whether HRF serves as a mediator in these pathways. The participants were 342 children (156 girls; Mage = 8.40, SD = 0.50) recruited in Texas. Children’s FMS (locomotor and ball skills) were assessed. School-based PA that included light, moderate, and vigorous PA was captured by accelerometers. The FITNESSGRAM battery was used to measure children’s HRF, including body composition, cardiorespiratory fitness, and muscular fitness. Structural equation models were used to evaluate two proposed models (model-1 = FMS»HRF»PA; model-2 = PA»HRF»FMS). Both locomotor and ball skills were associated with all components of HRF (*p* < 0.01), but not PA. The SEM analyses supported associations between FMS, HRF and PA, with sound goodness-of-fit indices: (1) model-1: CFI = 0.95; RMSEA = 0.072; and (2) model-2: CFI = 0.95; RMSEA = 0.071, respectively. The relationship between FMS and PA was fully mediated by the HRF in both directions. The behavioral mechanism (e.g., maintaining appropriate levels of HRF) provides meaningful insights to understand the obesity trajectory during middle childhood.

## 1. Introduction

It was well-documented that physical activity (PA) contributes to overall health and prevents childhood obesity, however, more than 70% of the children aged 6–11 years in the United States do not meet the recommended levels of PA, and less than half of the youth achieve adequate fundamental motor skills (FMS) and health-related fitness (HRF) levels [1,2,3,4]. Physical activity and physical fitness are closely related, in that physical fitness is mainly determined by physical activity patterns over weeks or months. FMS is defined as the degree of skill performance in goal-directed tasks, involving movement coordination (i.e., gymnastics) and manipulation of the movement patterns [5]. Specifically, a proficient level of fundamental motor skills is required for performance in a variety of daily physical activities. There is a significantly higher drop rate of PA levels in children aged 6–11 years (42.5% drop rate) than in adolescents aged 12–15 and 16–19 years of age (7.5% and 5.1%, respectively)—these percentages reflect the number of youth meeting PA recommendations measured by the accelerometer in each group [4]. A review of research indicated that physical inactivity and insufficient FMS competence contribute to the trajectory of obesity in early, middle and later childhood [5,6,7]. Habitual physical inactivity and low HRF trajectories are more likely to cluster together with the obesity epidemic, however, the underlying behavioral mechanism is not clear in the current literature, especially during childhood [8].

To effectively combat the obesity epidemic, investigating the potential causal relationships between PA behavior and FMS competence or the mediators of this association may show a positive spiral of engagement identified in a conceptual theoretical foundation [7,9]. Stodden and colleagues [7] proposed this conceptual model hypothesizing a reciprocal and developmentally dynamic relationship between FMS competence and PA, as well as a mediating role of HRF in this emergent relationship. The conceptual model places an emphasis on how the interaction of these behavioral factors may impact the trajectory of childhood obesity in early, middle and later childhood [7]. FMS, consisting of locomotor and manipulative skills, are cornerstones for engaging in regular PA and promoting HRF [5,6,7]. FMS competence also lays a foundation for engagement in organized and non-organized PA in children and adolescents [8,10,11,12,13,14]. For example, a recent intervention study [13] reported that a playful and highly varied physical activity program was effective for the development of preschoolers’ motor proficiency. Wrotniak et al. [14] also found that children in the top quartile of motor skill proficiency spent a significantly higher amount of time in moderate-to-vigorous PA compared to children with lower levels of motor skill proficiency. Furthermore, children aged 9–10 years old with adequate motor skill proficiency were more likely to participate in PA than their counterparts with poor motor skill proficiency in later childhood [15]. Overall, FMS competence forms the foundation for children’s physical and motor development and determines their choice to remain physically active in later childhood years [16,17].

Recent systematic reviews have also documented a positive relationship between FMS competence and HRF in children and adolescents [6,18,19]. For example, Cattuzzo and colleagues [18] reported that there was a consistent positive association of FMS competence with cardiorespiratory and muscular fitness, and a strong inverse association with body weight status in children and adolescents. A five-year longitudinal study found that children who demonstrated higher FMS competence scored consistently and significantly higher on a cardiovascular fitness test than their counterparts who showed lower FMS competence over a period of five years [12]. Similarly, it was also reported that children with high FMS competence performed significantly better than children with low FMS competence on all nine fitness test items at both pre- and post-tests (after 32 months) [20]. Interestingly, childhood manipulative skill competence, but not locomotor skills, was significantly associated with their cardiovascular fitness in adolescence [8]. Considering the rapid declines in HRF and PA in children as they develop into adolescence, understanding the behavioral mechanism and relationships among these variables (i.e., PA, HRF, and FMS) is extremely desirable from a public health perspective [4].

Given the increasing trend in severe obesity among U.S. youth, initiatives of early detection and targeted interventions should be of high priority, since the early school years are the most cost-effective time to intervene [21]. It is crucial to develop healthy movement behaviors [22,23], such as engaging in regular PA, sitting less, and developing sufficient FMS competence as part of children’s development. To our knowledge, few studies have tested the hypothesized associations among FMS, HRF and PA represented in Stodden’s model [7] in middle childhood. Gaining a better understanding of developmental mechanisms influencing PA trajectories in middle childhood may offer strategies to prevent the decline in PA and HRF typically observed during later childhood [15]. The main purpose of this study was to examine relationships among FMS competence, HRF, and PA participation in school-aged children using reliable and validated grade-level standard-based PE Metrics^TM^ [24], the FitnessGram test [25], and an objective measure of PA. The second purpose attempted to investigate the extent to which HRF components mediated the relationship between FMS competence and PA participation during middle childhood.

## 2. Materials and Methods

The study protocol was approved by the university institutional review board (IRB No.16-193), and the support and approval were also granted by the school district, principals and physical education (PE) coordinators. Parental informed consent forms were obtained in accordance with the participating school district and the Declaration of Helsinki before the data collection.

### 2.1. Participants

This study took place in four public elementary schools selected from the same school district in North Texas in the U.S. Initially, 403 third grade children were recruited. Sixty-one (61) children were removed from the data analysis because more than three days of PA measurement and/or more than two skill assessments were missed. The remaining 342 children (8.4 ± 0.50 years, 54% boys) were included in the data analysis. The race/ethnicity distribution was 31.8% White, 37.6% Hispanic, 19.9% African American, 4.9% Asian American, and 5.8% “others”. According to the Income Eligibility Guidelines (IEGs), 46% of the children were enrolled in the free/reduced lunch program.

### 2.2. Measures

#### 2.2.1. Demographic Information

Participants’ age, gender, and race/ethnicity information were obtained from the school district to characterize the sample. The socioeconomic status (SES) was identified based on each participant’s meal status (i.e., paid, free or reduced lunch) according to the IEGs.

#### 2.2.2. Body Mass Index (BMI)

Children’s height and weight (without shoes) were measured by using the Health-O-Meter^®^ Digital Scale. Height was measured to the nearest 0.1 cm and weight was measured in minimal clothing and to the nearest 0.1 kg. Body mass index (BMI) was calculated (kg/m^2^) and BMI percentiles were determined based on age and gender using the Centers for Disease Control and Prevention growth charts.

#### 2.2.3. Fundamental Motor Skills (FMS)

Children’s FMS were assessed by using PE Metrics^TM^ [24] in two categories: (1) locomotor skills: hopping and sliding; and (2) manipulative skills: underhand throwing and dribbling with hand. The PE Metrics^TM^ is a valid and reliable national standards-based assessment tool for measuring school-aged children’s FMS, developed by the NASPE’s assessment Task Force [26]. All FMS assessments were scored by two trained researchers. Before the assessment took place in the schools, research assistants completed three days of training in a university laboratory to ensure adequate qualifications. Two field assessment trials were arranged to reach a minimum of 95% inter-observer agreement for all four skills. According to the four-level scoring rubric (4 = Consistently; 3 = Usually; 2 = Sometimes; 1 = Seldom), the form and consistency of action of the child’s performance for dribbling, hopping, and sliding were scored separately, and each skill score ranged from 0 to 8. For underhand throwing, children performed three trials, and the total score of the three trials was used for the analysis (ranging from 0 to 24). The composite score for locomotor skill (sliding and hopping; values ranging from 0 to 16) and manipulative skill (underhand throwing and dribbling with hand; values ranging from 0 to 32) were used in the data analyses.

#### 2.2.4. Health-Related Fitness (HRF)

The HRF components were measured by FITNESSGRAM test battery [25]. Specifically, cardiorespiratory fitness was assessed by the Progressive Aerobic Cardiovascular Endurance Run (PACER), based on the total number of successful PACER “laps” that the students completed. Muscular fitness, including UpStrength and AbStrength, was measured by curl-ups and the 90-degree push-ups tests [27]. Body composition was represented by body mass index (BMI) using the standard formula: BMI = weight (kg)/[height (m)]^2^. BMI percentiles were determined based on age and gender using the U.S. Centers for Disease Control and Prevention growth charts for all participants [28]. The raw scores were used for all fitness components in the structural equation modeling (SEM) testing as latent variables.

#### 2.2.5. School-Based Physical Activity (PA)

Children’s school-based light PA (LPA), moderate PA (MPA) and vigorous PA (VPA) were objectively measured using Actical accelerometers. Children wore the accelerometer (Actical monitors; Mini-Mitter Co., Inc., Bend, OR, USA) on their non-dominant wrists for five consecutive school days from 8:00 a.m. to 3:00 p.m. (~7 h). Total time for LPA, MPA, and VPA were averaged by the days of wearing the accelerometers. When collecting the accelerometer data, research assistants visited each classroom and distributed the numbered accelerometers to ensure each child wore the same numbered accelerometer over the five days. The daily accelerometer log was used to track the wearing time for each participant, and the minimum wearing time (≥5 h; Mean = 391 min, SD = 15.8) during the school day was used for the PA data analysis. In order to capture the spontaneous activities of children, the accelerometers were initialized to save data in 60-s intervals (epochs), after taking into account age, gender, height, and weight [29,30]. Standard cut-points based on activity energy expenditure (AEE) for MVPA using ActiCal accelerometers were adopted to determine MVPA time (light-intensity as 0.01  ≤  AEE  <  0.04  kcal/kg/min; moderate-intensity as 0.04  ≤  AEE  <  0.10  kcal/kg/min; and vigorous-intensity as AEE  ≥  0.10  kcal/kg/min).

### 2.3. Statistical Analysis

Data were analyzed using IBM SPSS 26.0 Statistics for Windows and AMOS Version 26.0. For the descriptive analysis, we computed statistics including mean and standard deviation for all the dependent variables: FMS (locomotor and ball skills), school-based LPA, MPA, and VPA, and all components of HRF (see Table 1). A multivariate analysis of covariance (MANCOVA) analysis was conducted to examine the gender and ethnic differences after controlling for age. Pearson–product moment correlation was used to examine the bivariate correlations among the study variables (see Table 2). The hypothesized associations between PA and FMS, and the mediational role of HRF in this relationship were tested based on the conceptual model proposed by Stodden and colleagues [7]. Specifically, the first SEM model was structured to test the direct and indirect effects simultaneously of FMS on students’ PA through all components of HRF. Then, the second SEM model was structured to test the reverse pathway from PA to FMS by testing the mediation effect of HRF. The model fit was determined using goodness-of-fit statistics [31,32], including chi-square goodness-of-fit test (χ^2^/df < 5.0); comparative fit index (CFI); incremental fit index (IFI); normed fit index (NFI); and root-mean square error of approximation (RMSEA).

## 3. Results

The results for descriptive statistics (means and SDs) are presented in Table 1. Over 60% of the children did not reach the recommended “competent level-3” or above in overall FMS competence according to the PE Metrics™ criteria [24] with 63.9% in sliding, 69.5% in dribbling with hand, 76.8% in throwing, and 72.7% in hopping. Forty-eight percent (48%) of the children were in the overweight/obese category with the BMI percentile ≥ 85%. In this age group, MANCOVA indicated that boys participated in more MVPA than girls (*p* < 0.001; η^2^ = 0.06), and there were no significant age and ethnic effects on the study variables.

Based on the correlation analyses (see Table 2), all components of HRF and FMS (locomotor and ball skills) were significantly associated with one another (rs ranging from 0.13 to 0.28, *p* < 0.05). LPA was significantly related to curl-up test (r = 0.19, *p* < 0.01), and VPA was significantly and positively associated with the PACER and push-up tests (r = 0.12, r = 0.13; *p* < 0.01, respectively).

### Associations between FMS, HRF and PA

According to the first SEM model (FMS»HRF»PA; see Figure 1), the goodness-of-fit indices suggested a well-fitting model (χ^2^/df = 63.79/23 < 3; NFI = 0.92; IFI = 0.95; CFI = 0.95; RMSEA = 0.072; 90% CI [0.05, 0.09]). FMS explained 17% of variance in HRF (*p* < 0.01) and both locomotor and ball skills accounted for the variance explained in HRF. The direct effect between FMS and HRF (βlocomotor = 0.25, βball = 0.29; *p* < 0.001; respectively) and between HRF and PA (β = 0.25, *p* < 0.01) were significant and stronger than that of between FMS and PA. The model fit indicated that FMS was fully mediated through HRF components to PA, since the direct effects between FMS and PA were not significant in the first model.

According to the second SEM model (PA»HRF» FMS; see Figure 2), the goodness-of-fit indices suggested a good fit model (χ^2^/df = 64.74/24 < 3; NFI = 0.92; IFI = 0.95; CFI = 0.95; RMSEA = 0.071; 90% CI [0.05, 0.09]). PA explained 4% variance in HRF (*p* < 0.001) and all components of HRF significantly contributed to the variance in both locomotor and ball skills (Rlocomotor = 0.11, R = ball = 0.12; *p* < 0.001; respectively). The direct effect of HRF on FMS (βlocomotor = 0.33, β = ball = 0.35; *p* < 0.001; respectively) was significant and stronger than the direct effect of PA on FMS (β = 0.21; *p* < 0.001). The direct effect between PA and FMS was not significant in the second model, which suggests that the pathway from PA to FMS was fully mediated by all components of HRF.

## 4. Discussion

The results in this study supported the hypothesized directional pathways proposed by Stodden and colleagues [7] among FMS competence, HRF and PA during middle childhood (average nine years old). The major aims of the present study were to investigate the associations between FMS, HRF and PA; and to test the potential mediating effect of all components of HRF in this relationship, guided by Stodden’s conceptual model [7]. A growing body of literature suggests that FMS competence is an underlying mechanism potentially driving PA, HRF, and preventing obesity [33]. The findings of this study show that FMS is central to the understanding of PA and obesity in middle childhood, which indicates that the relationship between FMS and PA is not direct but mediated by HRF.

It is important to mention that most children in our study did not meet recommended grade-level competence in FMS (63.9% in sliding, 69.5% in dribbling with hand, 76.8% in underhand throwing, and 72.7% in hopping). This is consistent with findings showing that 77% of young children demonstrated delays in FMS [34]. Previous research has also reported that only 11% of adolescents (12- and 13-year-olds) demonstrate mastery or near mastery in a combination of nine FMS [35]. This is an alarming fact—children that show FMS delays in childhood may consistently demonstrate some delays in adolescence, which may prevent them from making a successful transition towards more advanced sport participation. In the present study, children were, on average, nine years of age and in the important transition period from early to later childhood. However, the majority of the children did not develop sufficient FMS competence (probably not enough instruction, practice, and experience to improve their competence), indicating that motor skill intervention is essential to reduce the risk of delays and further FMS development [10,36,37]. These findings provide important implications for practitioners; that is, quality school PE programs should strive to achieve all national standards, with particular attention to skill competency (standard 1) and valuing physical activity (standard 5). School-aged children should be provided with sufficient opportunities and positive experiences for learning and mastering various motor skills during a school PE program, as these may correlate to their PA participation and fitness promotion from middle childhood to adolescence [8,16].

The central finding of this study revolves around the antecedent-consequent behavioral mechanism of obesity trajectory in middle childhood. Several previous studies demonstrated the direct associations of FMS with PA and HRF in early childhood (i.e., among preschoolers) [10,38,39] and in later childhood [15,40], respectively. However, the mediating role of HRF in the bidirectional relationship between PA and FMS proposed in Stodden et al.’s conceptual model has not been tested during middle childhood. For example, a previous study [10] supports a direct association from FMS to PA among a group of kindergarteners (Mage = 5.37, SD = 0.48), which directly contributes to young children’s sedentary behavior in later school years. In investigating 565 fourth grade (9–10 years old) students’ FMS competence and HRF components [40], it was found that manipulative skills significantly predicted PACER, push-up, and trunk lifts tests, but not the curl-up test (for both boys and girls). Boys and girls in the skill-competent group significantly outperformed their counterparts on PACER, push-up, and trunk lifts tests, again, with the exception of curl-up test [40]. The study indicated that students who were competent in manipulative skills demonstrated a higher and healthy level in cardiovascular endurance, upper-body muscular strength and endurance, and flexibility. A recent study [15], focused on fourth and fifth graders (Mage = 10.87, SD = 0.77), also confirmed that regardless of gender, children with low FMS competence demonstrated lower in-class PA (step counts) and lower cardiorespiratory fitness, compared with children with the higher FMS competence. Although there were no statistically significant gender and ethnic differences on both locomotor and ball skills, it is worth noting that the significant gender difference on MVPA was found in this third grade sample, which may be a potential reason for the low fitness and FMS later found when those children moved up to fourth and fifth grades, especially among girls. PA participation and fitness promotion have been emphasized to fight the obesity epidemic in school-aged children; likewise, developing and mastering FMS should become a core part of the curriculum in elementary school PE, as FMS provides a foundation for cognitive, social, and physical growth [36].

During third grade (around 9 years of age) is the first time that children are administered HRF testing in the U.S., thus, the findings of this study provide insights for school-based PE curriculum design and PA/PE program implementation. Specifically, enhancing skill learning and PA participation during school (i.e., recess, classroom brain breaks) should be emphasized, and a fitness routine should be implemented into a daily PA/PE program. Another variable interacting with the relationship between FMS and PA we explored was weight status (BMI). Studies hint at a strong relationship between weight status, FMS, and other variables. For example, children with higher HRF and FMS competence are less prone to overweight or obesity during childhood [41]. From another perspective, children who are overweight/obese were found to have lower FMS competence [42]. These findings highlight that the associations between these variables may depend on the age, gender, race/ethnicity, and other biological factors of the children and need further longitudinal investigation. Nevertheless, promoting FMS, HRF, and PA in children is an essential component of their present and later weight status.

Limitations of this study included that the FMS was assessed with only four different locomotor and ball skills. Children’s lack of familiarity with the testing procedures may have affected their performance (such as doing a push-up in the Fitnessgram) [43]. In addition, it should be noted that one limitation of the study was the impossibility of establishing causal relationships between FMS, HRF and PA due to the nature of cross-sectional data. Future longitudinal studies are warranted to explore the temporal order of these variables and the potential mediating role of HRF. However, the findings outweigh the limitations by providing great insights into the understanding of these relationships in middle childhood. This stage of child development is important to the health risk behavior of physical inactivity among children. Early prevention appears to be rooted in the development of FMS, HRF, and PA.

## 5. Conclusions

In conclusion, the findings of this study demonstrated that the relationship between FMS and PA is not direct, but mediated by HRF among third grade children. These findings suggest that maintaining appropriate levels of HRF may benefit skill acquisition and promote PA in middle childhood. Taken together, these findings will assist in understanding the underlying mechanisms of behavioral change in order to strengthen effective intervention components and eliminate or adapt ineffective components. However, further research is needed to replicate the importance of HRF as a mediator between FMS and children’s engagement of different intensity levels of PA. This study provides empirical evidence for the inclusion of FMS development/acquisition as a mechanism of behavioral change and, thus, a strategy that should be included in interventions aimed at increasing children’s PA and HRF by age, gender, and race/ethnicity group.

## Figures and Tables

**Figure 1 children-08-00064-f001:**
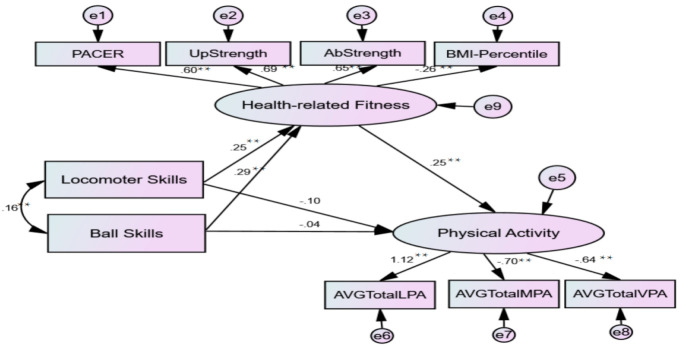
The Structural Model with Standardized Path Coefficients Model-1: FMS»HRF»PA. PACER = Progressive Aerobic Cardiovascular Endurance Run; PA = physical activity; BMI = body mass index. * *p* < 0.05; ** *p* < 0.01.

**Figure 2 children-08-00064-f002:**
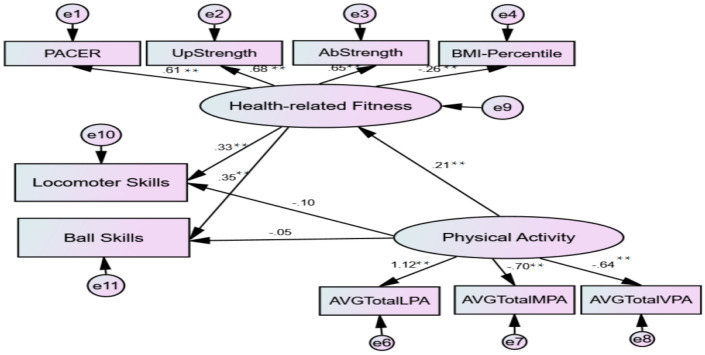
The Structural Model with Standardized Path Coefficients for Model-2: PA»HRF»FMS. PACER = Progressive Aerobic Cardiovascular Endurance Run; PA = physical activity; BMI = body mass index. * *p* < 0.05; ** *p* < 0.01.

**Table 1 children-08-00064-t001:** Sample Characteristics and Descriptive Statistics (N = 342).

Study Variables	Min	Max	Boys M (SD)	Girls M(SD)
BMI-percentile	3	97	74.44 (25.18)	70.26 (27.74)
PACER	3	73	20.92 (12.51)	18.26 (10.84)
Upper body strength	0	34	9.70 (6.87)	7.21 (5.5.0)
Abdominal strength	0	75	17.19 (17.03)	18.52 (17.33)
Total wearing time	325.8	429.2	391.43 (15.53)	391.52 (16.25)
Light PA	96	334	266.88 (33.53)	280.25 (40.44)
Moderate PA	23	210	92.09 (27.79)	84.13 (31.78)
Vigorous PA	0	103	16.50 (14.13)	10.21 (11.03)
Fundamental motor skills				
Ball skills	3	29	20.18 (3.17)	18.29 (3.21)
Locomotor skills	0	14	8.71 (2.45)	9.13 (2.25)

M = mean; SD = standard deviation; PACER = Progressive Aerobic Cardiovascular Endurance Run; PA = physical activity; BMI = body mass index. Total Wearing Time = Minutes.

**Table 2 children-08-00064-t002:** Pearson Correlation Matrix of the Variables (N = 342).

Variables	1	2	3	4	5	6	7	8	9
1. BMI-percentile	1								
2. PACER	−0.19 **	1							
3. Upper body strength	−0.24 **	0.38 **	1						
4. Abdominal strength	−0.12 *	0.39 **	0.47 **	1					
5. Locomotor skills	−0.17 **	0.28 **	0.16 **	0.13 **	1				
6. Ball skills	0.05	0.23 **	0.26 **	0.19 **	0.15 **	1			
7. Light PA	0.04	0.06	0.09	0.19 **	−0.03	0.02	1		
8. Moderate PA	0.03	−0.05	−0.07	−0.19	−0.04	−0.02	−0.78 **	1	
9. Vigorous PA	−0.10	0.12 *	0.13 **	0.07	0.10	0.03	−0.71 **	0.47 **	1

M = mean; SD = standard deviation; BMI = body mass index. * *p* < 0.05; ** *p* < 0.01.

## Data Availability

The data presented in this study are available on request from the corresponding author. The data are not publicly available due to ethnical restrictions.

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
