# Peer review of "A Mediation Analysis of the Association between Fundamental Motor Skills and Physical Activity during Middle Childhood"

_children, 2021, doi:10.3390/children8020064_

Round 1
Reviewer 1 Report
The study explores the relationship between Fundamental Motor Skills (FMS), Physical Activity (PA), and Health-Related Fitness (HRF) on a sample of middle childhood participants (6-8 years). The article is well written, structured, and quite interesting. The paper is potentially useful for reducing obesity rates among youth.
The introduction is quite clear but could be improved. The authors have based the introduction on previous literature. However, they are encouraged to add some more recent references (2016-2020). The materials and methods are, in general, well-described and organized. The results are clear but the discussion needs a review.
Specific comments
[Page 3, line 105] Please carefully check the English language. The title of section 2.2.1 should be “Demographic Information”.
[Page 4, line 151] The citation [7] should be between square brackets, following the journal guidelines.
[Page 6, lines 199-201] Please delete the three first lines since they are part of the journal template but not part of the manuscript.
[Discussion] Firstly, The authors discuss their results with previous studies. Throughout the discussion, several groups of age are mentioned (e.g. kindergarteners, third grade, fourth grade, fifth grade, etc.). I believe the discussion would be clearer for a potential reader if estimated ages are added. This would provide more clarification to the work made by the authors. Secondly, the authors have clearly stated that in previous work, the results are similar regardless of gender. However, what happens in the current manuscript? I think a clearer idea of the differences/similarities according to gender, age, and race/ethnicity could be added.
[References] Not all the references meet the journal requirements. For example, following the journal guidelines, only the volume should be provided when referencing journal articles (not the number of the article). Additionally, the use of italics should be reviewed.
The effort made by the authors is appreciated and the comments are expected to be useful.
Author Response
Response to Reviewers’ Comments
Manuscript ID: children-1057265
Manuscript Title: Understanding the Trajectory of Childhood Obesity: A Reciprocal Pathway between Fundamental Motor Skills and Physical Activity
Revised title: ”Reciprocal Pathways between Fundamental Motor Skills, Health-related Fitness, and Physical Activity during Middle Childhood”.
Reviewer-1
The study explores the relationship between Fundamental Motor Skills (FMS), Physical Activity (PA), and Health-Related Fitness (HRF) on a sample of middle childhood participants (6-8 years). The article is well written, structured, and quite interesting. The paper is potentially useful for reducing obesity rates among youth.
The introduction is quite clear but could be improved. The authors have based the introduction on previous literature. However, they are encouraged to add some more recent references (2016-2020). The materials and methods are, in general, well-described and organized. The results are clear but the discussion needs a review.
Response: Thank you for the favorable feedback. In this round of revision, we have carefully reviewed and addressed each reviewer’s comments. The introduction has been revised by adding most recent literatures and the reference 11, 12, and 13 have been replaced with most recent studies (see below).
- Jones, D.; Innerd, A.; Giles, EL.; Azevedo, LB.; Association between fundamental motor skills and physical activity in the early years: A systematic review and meta-analysis. J Sport Health Sci. 2020, 9, 542-552.
- Bolger, LE.; Bolger, LA.; O'Neill, C.; Coughlan, E.; O'Brien, W.; Lacey, S.; Burns, C.; Accuracy of children's perceived skill competence and its association with physical activity. J Phys Act Health. 2019, 16, 29-36.
- Gallotta, MC.; Baldari, C.; Guidetti, L.; Motor proficiency and physical activity in preschool girls: a preliminary stud Early Child Dev. Care. 2018, 188, 1381-91.
Specific comments
[Page 3, line 105] Please carefully check the English language. The title of section 2.2.1 should be “Demographic Information”.
Response: The typo has been corrected. Line 108.
[Page 4, line 151] The citation [7] should be between square brackets, following the journal guidelines.
Response: The format has been corrected. Line 156.
[Page 6, lines 199-201] Please delete the three first lines since they are part of the journal template but not part of the manuscript.
Response: The redundant information has been deleted. Line 205.
[Discussion] Firstly, The authors discuss their results with previous studies. Throughout the discussion, several groups of age are mentioned (e.g. kindergarteners, third grade, fourth grade, fifth grade, etc.). I believe the discussion would be clearer for a potential reader if estimated ages are added. This would provide more clarification to the work made by the authors.
Response: The specific age group has been identified and added in the text. Line 207, line 238, line 240, and line 246-247.
Secondly, the authors have clearly stated that in previous work, the results are similar regardless of gender. However, what happens in the current manuscript? I think a clearer idea of the differences/similarities according to gender, age, and race/ethnicity could be added.
Response: We conducted the additional multivariate analysis of covariance (MANCOVA) to examine the gender and ethnic differences after controlling for age. The data analysis, result and the discussion sections have been revised accordingly. Line 151-152; line 167-169, line 220-221, line 246-256, and Table 1.
[References] Not all the references meet the journal requirements. For example, following the journal guidelines, only the volume should be provided when referencing journal articles (not the number of the article). Additionally, the use of italics should be reviewed.
Response: The reference has been reviewed and corrected.
The effort made by the authors is appreciated and the comments are expected to be useful.
Response: Thank you for the thorough review. We hope we have successfully addressed the concerns in this round of revision. We look forward to hearing from you in the near future.

Reviewer 2 Report
Thank you for the opportunity to review this manuscript. Strengths include focus on an important health issue, a strong theoretical grounding, use of rigorous methods to assess outcomes, and a relatively large sample. Opportunities for improvement for the authors’ consideration are detailed below.
General
*Scattered typos were noted throughout. For example, I believe “few study has” in line 80 should be “few studies have” and lines 199-201 seem to be pasted directly from the author guidelines.
*The title frames this study as being about obesity trajectories, but the manuscript focuses on HRF, PA, and FMS. The role of BMI percentile is not discussed in much depth (including in the results). The authors may want to deepen the focus on obesity, including adding citations from the extensive research focusing on the role of fitness and PA in weight management, or consider changing the title.
Introduction
*Can the authors provide a clear definition of PA, HRF, and FMS. The difference between the concepts, while somewhat intuitive, may not be clear for all of this journal’s readership (which is general pediatrics).
*A visual representation of the Stoddard framework may be helpful to the reader, especially since Children is not a PA-focused journal and many readers may be unfamiliar.
Methods
*Lines 92 and 103 - The informed consent details are mentioned twice.
Results
*No comments.
Discussion
*Line 210-222 - This is important, and would benefit from being better fleshed out. The authors work and prior research demonstrates that the vast majority of children are not competent in FMS. What are the implications of this for research, clinicians, public health practitioners, and policymakers? Not all of those need to be addressed, but more depth and specificity about implications would be helpful.
*Line 223-239 - Can this paragraph be reworked? As it currently reads, it reports the results of prior studies but it does not put the current study into context. However how and why do the authors’ findings align with and differ from previous research?
Author Response
Response to Reviewers’ Comments
Manuscript ID: children-1057265
Manuscript Title: Understanding the Trajectory of Childhood Obesity: A Reciprocal Pathway between Fundamental Motor Skills and Physical Activity
Revised title: ”Reciprocal Pathways between Fundamental Motor Skills, Health-related Fitness, and Physical Activity during Middle Childhood”.
Reviewer-2
Thank you for the opportunity to review this manuscript. Strengths include focus on an important health issue, a strong theoretical grounding, use of rigorous methods to assess outcomes, and a relatively large sample. Opportunities for improvement for the authors’ consideration are detailed below.
Response: Thank you for the insightful review and feedback. We have carefully reviewed and addressed your comments. We look forward to hearing from you in the near future.
General
*Scattered typos were noted throughout. For example, I believe “few study has” in line 80 should be “few studies have” and lines 199-201 seem to be pasted directly from the author guidelines.
Response: We carefully proof read the manuscript and the relevant typos have been corrected accordingly. Line 85. Line 205.
*The title frames this study as being about obesity trajectories, but the manuscript focuses on HRF, PA, and FMS. The role of BMI percentile is not discussed in much depth (including in the results). The authors may want to deepen the focus on obesity, including adding citations from the extensive research focusing on the role of fitness and PA in weight management, or consider changing the title.
Response: The title has been revise to: ” Reciprocal Pathways between Fundamental Motor Skills, Health-related Fitness, and Physical Activity during Middle Childhood”.
Introduction
*Can the authors provide a clear definition of PA, HRF, and FMS. The difference between the concepts, while somewhat intuitive, may not be clear for all of this journal’s readership (which is general pediatrics).
Response: The brief definitions have been added in the introduction section. Line 36-40.
*A visual representation of the Stoddard framework may be helpful to the reader, especially since Children is not a PA-focused journal and many readers may be unfamiliar.
Response: We agreed with the reviewer that it will be helpful to include the original conceptual model (see below) from Stodden and colleagues. However, this may require the permission from Taylor & Francis (http://www.tandfonline.com) for the official publication/reprint in Children. We are open for this option—we will be happy to contact the relevant resource to get the permission if needed.
Methods
*Lines 92 and 103 - The informed consent details are mentioned twice.
Response: We have removed the repeated sentences in the participants section.
Results
*No comments.
Discussion
*Line 210-222 - This is important, and would benefit from being better fleshed out. The authors work and prior research demonstrates that the vast majority of children are not competent in FMS. What are the implications of this for research, clinicians, public health practitioners, and policymakers? Not all of those need to be addressed, but more depth and specificity about implications would be helpful.
Response: The implication has been added in the discussion. Line 226-231.
*Line 223-239 - Can this paragraph be reworked? As it currently reads, it reports the results of prior studies but it does not put the current study into context. However how and why do the authors’ findings align with and differ from previous research?
Response: Based on the comments from reviewer-1, we conducted additional data analysis to examine the group differences. So, we have added several discussion points in this paragraph. Line 246-256.
